# Transcriptome Characterization and Gene Changes Induced by *Fusarium solani* in Sweetpotato Roots

**DOI:** 10.3390/genes14050969

**Published:** 2023-04-25

**Authors:** Chengling Zhang, Qinchuan Luo, Wei Tang, Jukui Ma, Dongjing Yang, Jingwei Chen, Fangyuan Gao, Houjun Sun, Yiping Xie

**Affiliations:** Sweetpotato Research Institute, Chinese Academy of Agricultural Sciences, Xuzhou Institute of Agricultural Sciences in Jiangsu Xuhuai Area, Key Laboratory of Biology and Genetic Improvement of Sweetpotato, Ministry of Agriculture, Xuzhou 221131, Chinasunhouj1980@163.com (H.S.);

**Keywords:** *Fusarium solani*, gene, sweetpotato, transcriptome

## Abstract

Sweetpotato (*Ipomoea batatas*) is an important root crop that is infected by *Fusarium solani* in both seedling and root stages, causing irregular black or brown disease spots and root rot and canker. This study aims to use RNA sequencing technology to investigate the dynamic changes in root transcriptome profiles between control check and roots at 6 h, 24 h, 3 days, and 5 days post-inoculation (hpi/dpi) with *F. solani*. The results showed that the defense reaction of sweetpotato could be divided into an early step (6 and 24 hpi) without symptoms and a late step to respond to *F. solani* infection (3 and 5 dpi). The differentially expressed genes (DEGs) in response to *F. solani* infection were enriched in the cellular component, biological process, and molecular function, with more DEGs in the biological process and molecular function than in the cellular component. Kyoto Encyclopedia of Genes and Genomes (KEGG) pathway analysis showed that the main pathways were metabolic pathways, the biosynthesis of secondary metabolites, and carbon metabolism. More downregulated genes were identified than upregulated genes in the plant–pathogen interaction and transcription factors, which might be related to the degree of host resistance to *F. solani.* The findings of this study provide an important basis to further characterize the complex mechanisms of sweetpotato resistance against biotic stress and identify new candidate genes for increasing the resistance of sweetpotato.

## 1. Introduction

Sweetpotato (*Ipomoea batatas* (L.) Lam.) is the seventh most important food crop in the world and the fourth most important crop species in China. It is planted in more than 100 countries and is well known for its abundant nutritional value [1,2]. Sweetpotato is also widely served as a raw material in food and feed industries and as an energy crop. However, the production and quality of sweetpotato are significantly limited by *Fusarium*-infection-related diseases [3,4]. *F. solani*, with a wide range of hosts, is notorious for causing diseases in potato [5], medicago truncatula [6], bean [7], and other crops, causing root rot. *F. solani* can infect sweetpotatoes during the storage period, causing rot canker, and also infect seedlings and roots, causing root rot and canker in the field [3,8]. Irregular black or brown disease spots form on the sweetpotato seedlings and roots, leading to the death of the whole plant, with infected seedlings in serious cases. These spots can also lead to the formation of honeycomb cavities and a bitter taste in the root. If infected sweetpotatoes are stored, they can develop cellar rot and become rotten.

A large number of genes related to the defensive response to *Fusarium* spp. have been identified based on molecular findings. For example, *fhb1*, *fhb7*, and *TaWRKY70* genes in wheat confer resistance to *F. graminearum* infection [9,10,11]. Cotton infected by *Fusarium* spp. induces the expression of *GhMPK20*, *GbGSTU7*, and stearoyl-ACP desaturase *GhSSI2* significantly, whereas the silencing of *GhMPK20* and *GhSSI2* enhances resistance to *Fusarium* cotton wilt [12,13]. *GbGSTU7* positively regulates resistance to *Fusarium* wilt in *Gossypium barbadense*, whereas its silencing significantly reduces glutathione peroxidase activity in vivo and increases the incidence of *Fusarium* cotton wilt [13]. The *GhRLPGSO1-like* genes, including *GhRLP44*, *GhRLP6*, and *GhRLP34*, might play a role in resistance to *Fusarium* cotton wilt [14]. These genes are related to mitogen-activated protein kinase (MAPK), transcription factors (TFs), and glutathione transferases (GSTs) [12,13,14,15]. For more than a century, R genes have been identified in tomato (*Solanum lycopersicum*), such as *I*, *I-2*, and *I-3*, which are related to resistance to the *Fusarium* wilt caused by soil-borne fungus *F. oxysporum* f. sp. *lycopersici* [16,17,18].

Studies have also been conducted to identify the key genes induced by other pathogens or abiotic stress tolerance in sweetpotatoes [19,20,21,22,23]. *IbBAM1.1*, one of the β-amylase genes, acts as a positive regulator to enhance drought and salt stress resistance by regulating the level of osmoprotectants to balance osmotic pressure and activate the scavenging system to maintain reactive oxygen species (ROS) homeostasis in plants [22]. Additionally, *IbMYB308*, one of the R2R3-MYBs, and *IbPIF3.1*, one of the phytochrome-interacting factor genes, are positive regulators to enhance drought and salt stress resistance [20,23]. The over-expression of *IbPIF3.1, IbBBX24*, and *IbSWEET10* also significantly enhance *Fusarium* wilt tolerance in transgenic tobacco plants or *F. oxysporum* tolerance in transgenic sweetpotato [23,24,25]. A transcriptome analysis was carried out in sweetpotato in different developmental stages under abiotic or biological stress [26,27]. A large number of putative genes, such as *domain of unknown function 668* (*DUF668*), *chitin elicitor receptor kinase 1*, *ethylene-responsive transcription factor*, genes in MAPK, WRKY, NAC, MYB, GST, and protein kinase families, were found to be involved in the defense response against stress [26,27,28]. However, data on the genes involved in regulating the defense response to *Fusarium* infection in sweetpotato are still insufficient. Lin [28] identified various genes in sweetpotato that are differentially expressed during defense against *F. oxysporum,* which causes *Fusarium* wilt. The sweetpotato seedlings were used as materials, and only one infection time (24 hpi) was selected. The dynamic changes of genes in sweetpotato roots against *F. solani* were deficient. This study is conducted to elucidate the molecular mechanism of sweetpotato against *F. solani* at different infection times. RNA sequencing (RNA-seq) technology was used to investigate the dynamic changes in the transcriptome in nontreated sweetpotato roots and roots treated for 6 h, 24 h, 3 days, and 5 days (hour/day post-inoculation, hpi/dpi) of *F. solani*. Meanwhile, the differentially expressed genes (DEGs) in the plant–pathogen interaction and TFs in roots were also analyzed to understand how sweetpotato responds to the fungal infection. Further, the information obtained in this study might help improve the current understanding of plant–pathogen interactions.

## 2. Materials and Methods

### 2.1. Sweetpotato and Fungal Materials

*F. solani* was obtained from a decayed sweetpotato. The rDNA internal transcribed space (rDNA-ITS) and the small ribosomal subunit (SSU) were sequenced and deposited in GenBank under accession numbers KU341839 and KU356778, respectively. *F. solani* was cultured on sweetpotato dextrose agar containing 200.0 g diced sweetpotato, 15.0 g sucrose, 15.0 g agar, and 1.0 L of distilled water for 5 days at 28 °C. The culture plates were flooded with distilled water, and the spore suspensions obtained were adjusted to 10^6^ spores·mL^−1^ using a hemocytometer.

### 2.2. Inoculation of F. solani 

The sweetpotato (Xushu 32, a sensitive variety to *F. solani* ) roots were washed, dried naturally, disinfected with 75% alcohol, cut into 0.5 cm thick disks, and inoculated with 200 µL of spore suspension in the center using an Oxford cup. The infected sweetpotato disks were put into sterilized Petri dishes, with wet absorbent cotton added on the side to moisturize the disks, and incubated at 28 °C in an illumination incubator (Panasonic, MLR-352H-PC, Ehime, Japan) with 50% relative humidity for 5 days. Three replicates were established per treatment, and each replicate consisted of three roots. The thick disks without *F. solani* were used as a control check (CK). The samples were collected at 6 hpi, 24 hpi, 3 dpi, and 5 dpi, besides the CK. Three randomly selected individuals per time point and per type of treatment were used for further analysis. The disease severity was determined by the area of disease.

### 2.3. RNA Extraction, Library Preparation, and Illumina Sequencing

Total RNA from the nontreated samples (CK-32-1, CK-32-2, and CK-32-3) and 12 treated samples (T-6h-1, T-6h-2, T-6h-3, T-24h-1, T-24h-2, T-24h-3, T-3d-1, T-3d-2, T-3d-3, T-5d-1, T-5d-2, and T-5d-3) were extracted using the Omega Plant RNA Kit (Omega Bio-Tek, Norcross, CA, USA) following the manufacturer’s protocols, and the quality was checked with an Agilent 2100 bioanalyzer (Agilent, Palo Alto, CA, USA). The construction of cDNA libraries and RNA-seq analysis were performed by Genedenovo Bio-Tech Co., Ltd. (Guangzhou, China). The library was sequenced on an Illumina HiSeq 4000 platform with paired-end sequencing reads (2 × 100 bp).

### 2.4. Data Analysis

Raw sequences with adaptors, unknown nucleotides of more than 10%, and low-quality sequences containing more than 50% of low-quality (*Q* value ≤ 20) bases were filtered using fastp to obtain clean reads [29]. Short read alignment tool Bowtie2 [30] (version 2.2.8) was used for mapping reads to the ribosome RNA (rRNA) database. The rRNA-mapped reads were then removed. The remaining clean reads were further used in assembly and gene abundance calculation. The paired-end clean reads were mapped to the sweetpotato reference genome (http://sweetpotato.uga.edu/index.shtml, accessed on 30 January 2020) using HISAT2.2.4 (hierarchical indexing for spliced alignment of transcripts) [31], and the other parameters were set as default. For each transcription region, a fragment per kilobase of transcript per million mapped reads value was calculated to quantify its expression abundance and variations. Principal component analysis (PCA) was performed using the R package models (http://www.r-project.org/, accessed on 30 January 2020) in this experiment. The genes were considered significantly differentially expressed if the absolute log two-fold change was ≥2 and the false discovery rate was <0.05. The edgeR and DESeq2 [32] software were used to compare two samples or two different groups.

All DEGs were mapped to the Gene Ontology (GO) terms in the GO database (http://www.geneontology.org/, accessed on 2 February 2020). The gene numbers were calculated for every term, and significantly enriched GO terms in DEGs compared with the genome background were defined using the hypergeometric test. The KEGG pathway enrichment analysis identified significantly enriched metabolic pathways or signal transduction pathways in DEGs compared with the whole-genome background [33]. The GO terms and KEGG pathways with corrected *p*-values of ≤0.05 were considered significantly enriched.

### 2.5. Validations of RNA-seq Data Using Quantitative Real-Time Polymerase Chain Reaction

Quantitative real-time polymerase chain reaction (qRT-PCR) was performed to validate the RNA-seq results for four gene transcripts. Purified RNA (1 µg) was reverse transcribed to cDNA using the PrimeScript II First-Strand cDNA Synthesis Kit (TaKaRa, Beijing, China). The qRT-PCR was conducted on the Bio-Rad Real-Time PCR System (Bio-Rad, CFX96, Foster, CA, USA). The amplification program was as follows: 5 min at 95 °C, followed by 40 cycles of 95 °C for 15 s and 60 °C for 15 s. The qRT-PCR analysis was conducted with three technical replicates. The control, *IbActin*, was amplified using the primers Actin-F/R. All primers for qRT-PCR are listed in Table 1. The relative expression levels of the genes were calculated using the 2^–ΔΔCt^ method [34].

## 3. Results

### 3.1. Disease Severity, Transcriptome Generation, and Assembly

Conidia of *F. solani* were inoculated into the central disk of sweetpotato. After 2 days of *F. solani* infection, the disease symptoms were confined to the inoculated disk. Black spots extended on the inoculated surface on the third day, and the symptoms deteriorated on the fifth day. The 6 and 24 hpi time points, which point to the initial infection stage, were selected because of no or few symptoms. The 3 and 5 dpi time points, which point to the late infection stage, were also selected for RNA-seq due to obvious symptoms.

The number of reads per biological replicate for each time point is shown in Table 2. A total of 1083.66 million raw reads were generated for the samples. After a stringent filtering process, 1082.03 million clean reads remained. After quality control, the base content was more than 47%, and the quality of the sequenced bases Q20 and Q30 was more than 97% and 90%, respectively, indicating that the quality of the bases obtained by transcriptome sequencing was good and that they could be used for subsequent analysis (Appendix A).

The mapping percentage of CKs to the sweetpotato reference genome was more than 71%. The mapping percentages of 6 and 24 hpi to the sweetpotato reference genome were lower, while those to the *F. solani* reference genome were higher compared with those at the other times (Appendix A). Therefore, the pathogen was extremely active in infecting sweetpotatoes within 24 h.

A correlation analysis was performed using the R software. The correlation of two parallel experiments evaluates the reliability of experimental results and the operational stability. The correlation coefficient between two replicates was calculated to evaluate repeatability between samples. The higher the similarity of expression patterns between samples, the closer the correlation coefficient (*R*^2^) is to 1. As shown in Appendix A, the correlation coefficients between repeats were higher than 0.81, indicating that the results were reliable and the sample selection was appropriate. According to the heat map analysis and PCA (Figure 1), the plants could be divided into an early step (6 and 24 h) and a late step based on their response to *F. solani* infection (3 and 5 d).

### 3.2. Analysis of Sweetpotato DEGs in Response to F. solani

Four comparisons were made: CK-32 vs. T-6h (sweetpotato roots without *F. solani* (CK-32) relative to sweetpotato roots infected with *F. solani* at 6 h (T-6h)), CK-32 vs. T-24h (CK-32 relative toT-24h), CK-32 vs. T-3d (CK-32 relative toT-3d), and CK-32 vs. T-5d (CK-32 relative to T-5d). A total of 16,661, 12,459, 16,277, and 12,169 DEGs (ranging from 22.5% to 30.8% of total expressed genes) were detected in the CK-32 vs. T-6h, CK-32 vs. T-24h, CK-32 vs. T-3d, and CK-32 vs. T-5d comparisons, respectively. Among these, 4087, 4648, 5488, and 5253 DEGs were upregulated, whereas 12,574, 7811, 10,789, and 6916 DEGs were downregulated. Clearly, the numbers of downregulated genes were higher than those of the upregulated genes (Figure 2E).

### 3.3. Verification of the RNA-seq Results

qRT-PCR was used to amplify the five candidate DEGs at the five time points post-inoculation to validate the RNA-seq expression profiles of DEGs (Figure 3). Their expression showed an approximately linear correlation to the RNA-seq results.

### 3.4. GO and KEGG Analyses of the DEGs

The differentially expressed proteins were mapped to each term in the GO database, compared, and tested, and GO enrichment analysis was conducted on the DEGs. The results of the GO enrichment analysis showed that the DEGs involved in response to *F. solani* were enriched in the cellular component (CC), biological process (BP), and molecular function (MF).

Among the CK-32 vs. T-6h, CK-32 vs.T-24h, CK-32 vs. T-3d, and CK-32 vs.T-5d comparisons, the top genes in BP were related to the metabolic process, cellular process, and single-organism process; those in MF were related to transporter activity, catalytic activity, and binding, and those in CC were related to cell part, cell, and organelle (Table 2). More DEGs were assigned to the terms in BP and MF than to the CC terms (Table 2), while the percentages of DEGs related to CC, BP, and MP were similar. The top three terms in each domain, as listed in Table 2, were metabolic process, cellular process, and single-organism process in BP; catalytic activity, binding, and transporter activity in MF; and cell part, cell, and organelle in CC. Although these enriched terms were similar at different times after inoculation, the individual genes contributing to the common enriched terms were substantially diversified at different times after *F. solani* inoculation.

In organisms, genes usually interact with each other to play roles in certain biological functions. The pathway enrichment analysis identified significantly enriched metabolic pathways or signal transduction pathways in DEGs compared with the whole-genome background. The results of the KEGG analysis showed that 12,225 genes in CK-32 vs. T-6h, CK-32 vs. T-24h, CK-32 vs. T-3d, and CK-32 vs. T-5d were annotated and involved in 129–130 pathways. The top three pathways were metabolic pathways, the biosynthesis of secondary metabolites, and carbon metabolism (Table 3). Twenty pathways with the lowest *Q* values and the most significant enrichments were selected for display; the smaller the *Q* value, the more significant the enrichment (Appendix A).

### 3.5. Plant–Pathogen Interaction

Plants face a variety of pathogen infections during their development, and the two lines of defense in plants are generated to inhibit the destruction of pathogenics. The early stage of defense is the immune response pathogen-associated molecular pattern (PAMP)-triggered immunity (PTI), and the second line of defense is the immune response effector-triggered immunity (ETI). In the ETI, the plant directly or indirectly perceives pathogen effectors via resistance proteins and induces hypersensitive cell death at the infection site, which is named the hypersensitive response (HR) [35,36]. In the fungal PAMP pathway in this study, 18, 14, 24, and 13 DEGs (encoding the calcium-dependent protein kinase) that included 8, 8, 15, and 9 upregulated genes and 10, 6, 9, and 4 downregulated genes in the CK-32 vs. T-6h, CK-32 vs. T-24h, CK-32 vs. T-3d, and CK-32 vs. T-5d comparisons were identified, respectively. In the downstream reaction catalyzed by respiratory burst oxidase, 12, 12, 11, and 11 DEGs included 5, 5, 6, and 5 upregulated genes and 7, 7, 5, and 6 downregulated genes in CK-32 vs. T-6h, CK-32 vs. T-24h, CK-32 vs. T-3d, and CK-32 vs. T-5d comparisons, respectively. These genes might regulate the outbreak of ROS, thus causing the allergic necrosis of plants and cell wall reinforcement (Figure 4 and Appendix A).

In the calcium channel pathway, four types of DEGs were involved in regulation: (1) The cyclic nucleotide-gated channel: 7, 2, 5, and 2 downregulated genes and 4, 3, 3, and 1 upregulated genes were detected in CK-32 vs. T-6h, CK-32 vs. T-24h, CK-32 vs. T-3d, and CK-32 vs. T-5d comparisons, respectively; (2) calmodulin: 3, 2, 4, and 3 downregulated genes and 4, 5, 3, and 4 upregulated genes were detected in CK-32 vs. T-6h, CK-32 vs. T-24h, CK-32 vs. T-3d, and CK-32 vs. T-5d comparisons, respectively; (3) calcium-binding protein: 9, 5, 12, and 10 downregulated genes and 3, 3, 0, and 0 upregulated genes were detected in CK-32 vs. T-6h, CK-32 vs. T-24h, CK-32 vs. T-3d, and CK-32 vs. T-5d comparisons, respectively; (4) nitric oxide synthase in plants: 3, 2, 4, and 3 downregulated genes were detected in CK-32 vs. T-6h, CK-32 vs. T-24h, CK-32 vs. T-3d, and CK-32 vs. T-5d comparisons, respectively. These genes regulate stomatal closure and cell wall reinforcement (Figure 4 and Appendix A).

### 3.6. Expression of TFs

TFs are essential players in the regulatory networks that govern developmental processes and the deployment of pathogenicity factors during infection [37]. In the present study, 39 families of TFs and 510 genes encoding diverse putative TFs were DEGs induced by *F. solani*. Further, 35, 45, 45, and 43 putative TFs, which might be involved in the infection process, were upregulated in different stages of *F. solani* infection from 6 hpi to 5 dpi (Table 4).

## 4. Discussion

*F. solani* can infect sweetpotato in the seedling, field, and storage periods, causing stem decay and black rot in roots [3,38]. A series of reactions are induced by the pathogen in the host, causing resistance to the pathogen infection. RNA-seq can be used for the discovery of new genes, the identification of candidate gene families, transcription mapping, metabolic pathway determination, and evolutionary analysis [39,40]. The understanding of the complex physiological and molecular mechanisms induced by biological stresses, especially *Fusarium* infection, in sweetpotato roots remains limited [28,41]. In this study, RNA-seq was used to understand the interaction between *F. solani* and sweetpotato, and the data on sweetpotato was analyzed. Sample correlation analysis and PCA were in accordance with the symptoms induced by *F. solani* in sweetpotato host resistance to *F. solani* could be divided into an early step (6 and 24 hpi) and a late step (3 and 5 dpi), suggesting that the response at the transcriptional level in plants at 6 hpi is similar to that of plants at 24 hpi, whereas the response at 3 dpi is similar to that at 5 dpi.

DEGs in different pathways of the comparisons indicated that sweetpotato was challenged with *F. solani* via a complex network to stimulate plant immune system responses and regulate the expression of defense-related genes [26,28]. More than 12,000 DEGs were detected in CK-32 vs. T-6h, CK-32 vs. T-24h, CK-32 vs. T-3d, and CK-32 vs. T-5d comparisons, and these DEGs were enriched into different GO entries and KEGG metabolic pathways, reflecting that some genes in sweetpotato play an important role in the resistance to pathogens [42,43]. The number of downregulated genes was higher than that of upregulated genes. On the contrary, the numbers of upregulated genes were higher than those of downregulated genes in sweetpotato infected with *F. oxysporum* [28]. This might be related to different mechanisms of host resistance to different pathogens.

The most dominant GO terms identified during different durations of *F. solani* stress were metabolic process and cellular process in BP, catalytic activity and binding in MF, and cell and cell part in the CC response to stimulants. These results agreed with the findings [26] on hexaploid sweetpotato under salt stress but were not consistent with the findings on *F. oxysporum* infection [28]. In KEGG, more DEGs were found in metabolic pathways than in other pathways, implying that sweetpotato inoculated with *F. solani* starts the key mechanisms of chemical modifications, including the regulation of enzymes or kinase activity, and the synthesis and degradation of proteins or other substances such as S-(hydroxymethyl) glutathione or other mitochondrial substances [26].

Plants have gradually developed a complex immune system in the process of long-term interactions with pathogens [35,36]. ROS, which regulate the plant immune response and cause the HR of plants, are produced by calcium-dependent protein kinase and respiratory oxygen burst kinase (Rboh) in PTI [44]. In this pathway analysis, a large number of genes were differentially expressed, and the expression trends of some genes at different time points were consistent (Figure 4), which might play an important role in the resistance to *F. solani*.

The increase in Ca^2+^ concentration in the cytosol is also a regulator for the production of ROS and localized programmed cell death/HR [45]. Calcium-binding protein interacts with nitric oxide synthase (NOS) to generate NO and improves plant disease resistance by causing allergic necrosis [44,46]. In this pathway of research, the numbers of downregulated genes were higher than those of the upregulated genes. All genes were downregulated in NOS, which might be related to the resistant sweetpotato varieties to the disease, but this needs further experimental verification.

TFs are key regulatory factors that play important roles in plant biotic and abiotic stress resistance. Of these, WRKY TFs form one of the largest protein superfamilies in plants. They can regulate various defense processes and play important roles in controlling the transcription of defense-related genes by binding to W-Box cis-elements present in their promoters [47,48]. KEGG analysis showed that 43 WRKY genes were differentially expressed. Only 12 genes were upregulated at the four time points after *F. solani* inoculation. Of these, *WRKY45*/G217 and *WRKY75*/G45220 were significantly expressed compared with other genes at the four time points and strongly induced by *F. solani*. In the sweetpotato inoculated with *F. oxysporum*, a large number of WRKY genes were also significantly differentially expressed; these TFs have diverse biological functions and play important roles in the defense response [26,28]. As to the DEGs in CK-32 vs. T-24h compared with DEGs reported in the *F. oxysporum* infection of sweetpotato, we identified more WRKY genes in this study than before [28]. Additionally, *WRKY75*, *71* and *61* had similar expression trends in CK-32 vs. T-24h and XZH-F07 vs. XZH-CK (XZH, highly susceptible to *Fusarium* wilt). These genes may play an important function in broad-spectrum resistance to pathogens. Therefore, most DEGs were inconsistent. Sweetpotato is a hexaploid and has 90 chromosomes (2n = 6X = 90), with great homogeneity. The different varieties of sweetpotato or the different parts (seedling or root) may be related to different mechanisms against stresses [28,48]. The WRKY DEG number identified in sweetpotato seedlings in response to *F. oxysporum* infection was 8. These WRKY DEGs were different from our results.

The MYB family of proteins is large, functionally diverse, and represented in all eukaryotes [49]. The KEGG analysis showed that 57 MYB genes were differentially expressed, and only 10 were upregulated at the four time points after *F. solani* inoculation. Of these, *MYB4*/G20970 was significantly expressed compared with the other genes at the four time points, but *MYB2*/G22956 and *MYB2*/G40226 were significantly expressed at 3 and 5 dpi (late step) compared to 6 and 24 hpi (early step). Only one MYB gene, *IbMYB1-2b*, was involved in the defense response against *F. oxysporum* infection [28]. The TFs acted as transcriptional activators or repressors that regulated hormonal changes in plants not only during the early phases but also in the later step [46,48]. Another kind of TF was the basic helix–loop–helix (bHLH), the second largest TF family, which plays an important role in transcription and protein-interacting regulation in plants [50,51]. Nine bHLH genes in fifty DEGs were upregulated in this study. These results suggest that the TF proteins might function as key positive regulators or negative regulators in the sweetpotato defense against infection by *F. solani.* In the future, we will identify the disease resistance of sweetpotato varieties and screen the resistant varieties. The resistant variety transcriptome will be sequenced and analyzed to support the results or shortlist the true resistance genes.

## 5. Conclusions

In conclusion, plants thrive in certain environments and are continuously challenged by various forms of biotic stresses. The defense reaction of sweetpotato can be divided into an early step (6 and 24 hpi) and a late step based on their response to *F. solani* infection (3 and 5 dpi). The DEGs in response to *F. solani* were enriched in CC, BP, and MF. The DEGs were assigned to terms in BP and MF rather than CC terms. KEGG analysis showed that the top three pathways were metabolic pathways, the biosynthesis of secondary metabolites, and carbon metabolism. In the plant–pathogen interaction and TFs, more downregulated genes were identified than upregulated genes, which might be related to the degree of host resistance to *F. solani.*

## Figures and Tables

**Figure 1 genes-14-00969-f001:**
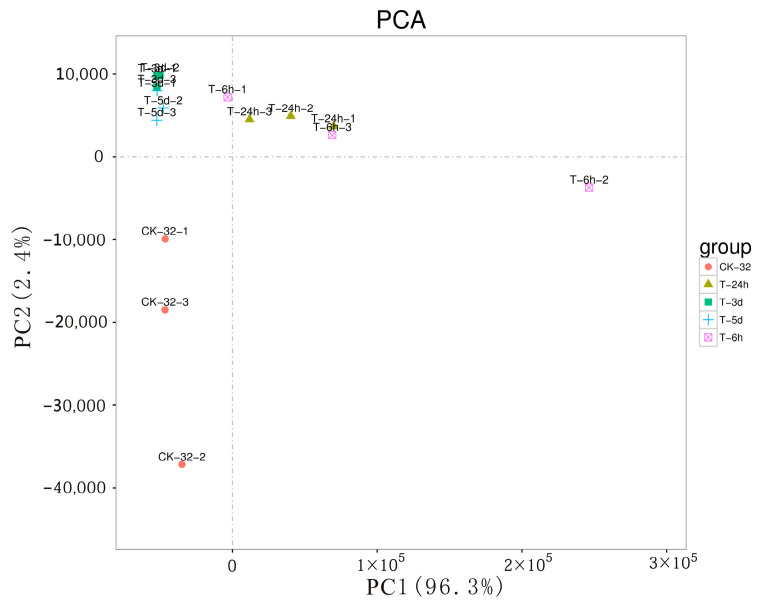
PCA (principal component analysis).

**Figure 2 genes-14-00969-f002:**
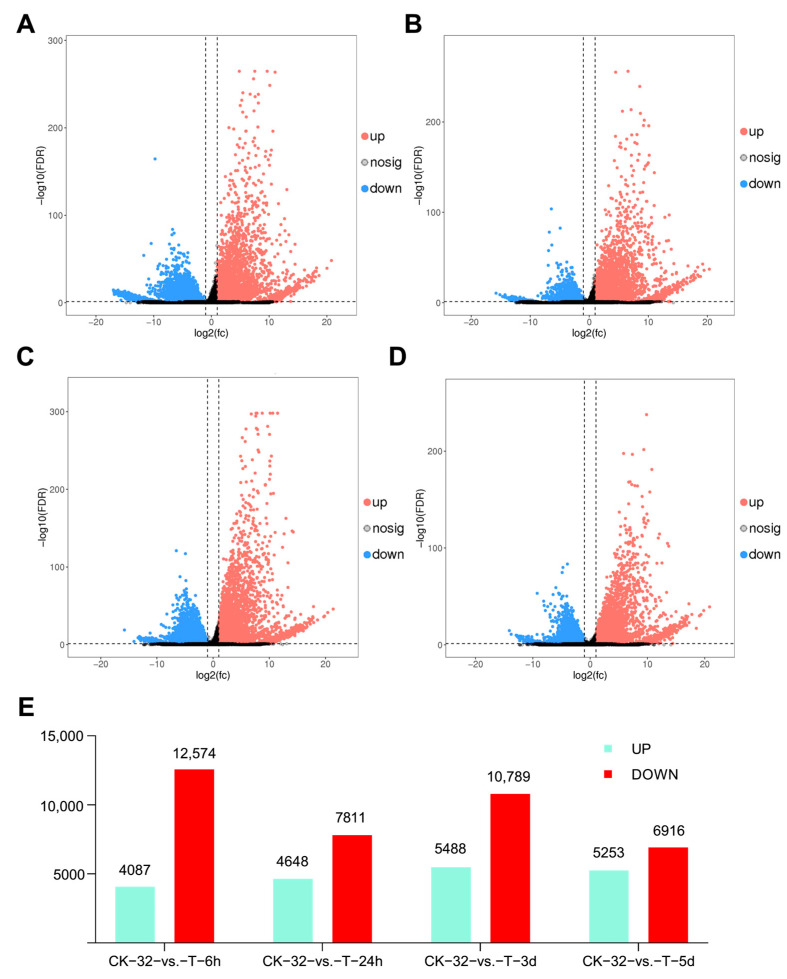
Clustering analysis of DEGs in the four comparisons. (**A**): CK−32 vs. T−6h; (**B**): CK−32 vs. T−24h; (**C**): CK−32 vs. T−3d; (**D**): CK−32 vs. T−5d; (**E**): differentially expressed genes (DEGs) in the four comparisons.

**Figure 3 genes-14-00969-f003:**
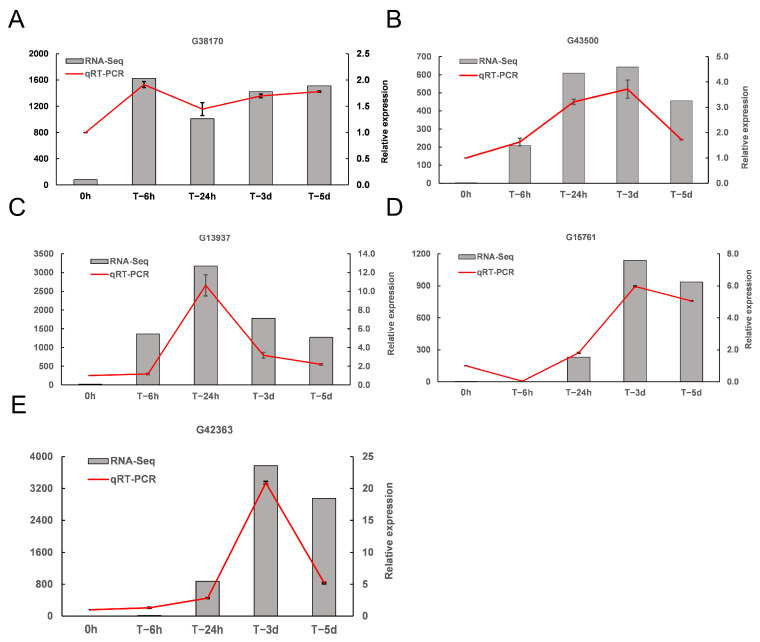
Expression of the selected four genes revealed by RNA-seq and qRT-PCR. (**A**): G38170; (**B**): G43500; (**C**): G13937; (**D**): G15761; (**E**): G42363. *Y*-axis (**left**) indicates RNA-seq data; *Y*-axis (**right**) indicates qRT-PCR data. Data from qRT-PCR are the means of three replicates, and bars represent SE. Data from RNA-seq are means of the replicates. *X*-axis indicates the treatments: 0 h, sweetpotato without *F. solani* infection; T−6h, sweetpotato at 6 h after *F. solani* infection (6 hpi); T−24h, 24 hpi; T−3d, 3 dpi; T−5d, 5 dpi.

**Figure 4 genes-14-00969-f004:**
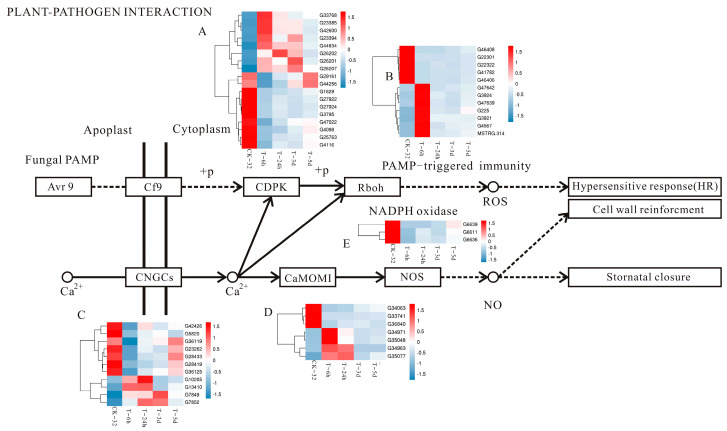
The partial plant–pathogen interaction pathway that involves differentially expressed genes, and the heat maps of genes in the plant–pathogen interaction pathway. (**A**): *calcium-dependent protein kinase genes* (CDBK); (**B**): *respiratory burst oxidase genes* (RBOH); (**C**): *cyclic nucleotide-gated channel genes* (CNGC); (**D**): *calmodulin* (CALM) and *calcium-binding protein genes* (CML); (**E**): *nitric oxide synthase genes* (NOS).

**Table 1 genes-14-00969-t001:** Primers used in this study.

Name	Sequence (5′–3′)	Name	Sequence (5′–3′)
Actin-F	AGCAGCATGAAGATTAAGGTTGTAGCAC	Actin-R	TGGAAAATTAGAAGCACTTCCTGTGAAC
qG38170-F	TCCAACTCTGAGCCGCCGCAGC	qG38170-R	TCCAAACTCTCCCCAATTAT
qG43500-F	ATGCAGGGGGCCATAAAACT	qG43500-R	ACTCAATAGCGCCTCCATCC
qG42363-F	CAAGGTCCCCGGAGGATGTAACA	qG42363-R	CTGAGGATAACTATAAGCATCAG
qG15761-F	GCCATCCTGAACCGGAGCGTCA	qG15761-R	CTGGTAAACAAGCTCCACCGTC
qG13937-F	TGTTTTGCTTGTGGTGGTGGCGC	qG13937-R	ACAGAAGGCAGTCCACCCATACT

**Table 2 genes-14-00969-t002:** Gene ontology classification analysis of DEGs.

Comparisons	Biological Process	Up	Down	Molecular Function	Up	Down	Cellular Component	Up	Down
CK-32 vs. T-6h	metabolic process	613	1831	catalytic activity	635	1584	cell part	269	952
	cellular process	546	1645	binding	440	1213	cell	269	952
	single-organism process	508	1303	transporter activity	81	161	organelle	182	731
	all	3022/9230		all	2920/8826		all	1722/5613	
CK-32 vs.T-24h	metabolic process	748	1051	catalytic activity	789	916	cell part	354	545
	single-organism process	588	787	binding	493	755	cell	354	545
	cellular process	641	999	transporter activity	87	115	organelle	244	437
	all	2253/9230		all	2233/8826		all	1301/5613	
CK-32 vs. T-3d	cellular process	756	1381	catalytic activity	935	1260	cell part	370	763
	metabolic process	869	1491	binding	576	998	cell	370	763
	single-organism process	711	1089	transporter activity	111	140	organelle	259	610
	all	2906/9230		all	1669/5613		all	2840/8826	
CK-32 vs. T-5d	single-organism process	707	702	catalytic activity	927	772	cell part	370	466
	cellular process	722	910	binding	536	627	cell	370	466
	metabolic process	853	960	transporter activity	115	98	organelle	267	379
	all	2236/9230		all	1249/5613		all	2195/8826	

Note: all: up/down.

**Table 3 genes-14-00969-t003:** Top 3 KEGG pathways in terms of representation of DEGs.

Comparisons	Pathway	CGPA (% of 12,225)	The Top 3 (DEGs Number/All Genes Number)		
CK-32 vs. T-6h	130	3728 (30.49%)	Metabolic pathways (1800/5181)	Biosynthesis of secondary metabolites (1207/3087)	Carbon metabolism (267/683)
CK-32 vs. T-24h	130	2766 (22.63%)	Metabolic pathways (1391/5171)	Biosynthesis of secondary metabolites (945/3087)	Carbon metabolism (223/683)
CK-32 vs. T-3d	129	3562 (29.14%)	Metabolic pathways (1843/5171)	Biosynthesis of secondary metabolites (1229/3087)	Carbon metabolism (282/683)
CK-32 vs. T-5d	129	2732 (22.35%)	Metabolic pathways (1493/5171)	Biosynthesis of secondary metabolites (1041/3087)	Carbon metabolism (240/683)

Note: CGPA: candidate genes with pathway annotation; the number 12,225 represents all genes with pathway annotations.

**Table 4 genes-14-00969-t004:** Differential expression of transcription factors induced by *F. solani* in sweetpotato.

Transcription Factor	CK-32 vs. T-6h	CK-32 vs. T-24h	CK-32 vs. T-3d	CK-32 vs. T-5d
AP2	2	2	4	3
ARF	8	6	8	7
ARR-B	3	1	1	1
B3	1	2	2	1
BBR	1	1	1	1
BES1	3	2	3	4
bHLH	29	23	32	27
bZIP	7	8	9	6
C2H2	16	11	11	10
C3H	8	3	3	2
CAMTA	0	1	0	0
CO-like	1	0	0	0
CPP	1	1	1	0
DBB	3	0	3	0
Dof	7	3	7	4
E2F/DP	2	1	1	2
ERF	5	4	4	5
FAR1	3	0	1	0
G2-like	9	6	11	10
GATA	5	2	6	5
GRAS	4	1	1	1
GRF	0	0	1	0
HB-other	6	3	2	1
HD-ZIP	15	10	13	10
HSF	8	7	7	7
LBD	1	1	4	3
M-type	1	0	1	0
MIKC	0	1	2	2
MYB	22	20	28	25
NAC	18	17	20	17
NF	4	3	3	3
Nin-like	0	0	1	0
SBP	5	3	4	3
STAT	0	0	1	0
TALE	7	6	6	5
TCP	4	1	2	2
Trihlix	4	2	3	2
Whrly	0	0	1	0
WRKY	17	21	24	17
Upregulated	35/230	45/174	45/232	43/188

## Data Availability

Not applicable.

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
