# Peer review of "Transcriptome Characterization and Gene Changes Induced by Fusarium solani in Sweetpotato Roots"

_genes, 2023, doi:10.3390/genes14050969_

Round 1

Reviewer 1 Report

The pathogenic fungi species Fusarium oxysporum and Fusarium solani cause fusarium wilt in sweet potato. The manuscript reports the transcriptome study on Fusarium solani infection of sweet potato root at different periods after inoculation.  More than 10,000 differentially expressed genes (DEGs) were identified. Based on DEG profiles, the process of F. solani infection could be divided into the early and late phases. Upregulation for the four selected genes was validated by qRT-PCR. DEGs of all KEGG pathways had more members downregulated, notably different from an earlier DEG report on Fusarium oxysporum infection of sweet potato (Lin et al. PLos One 2017).

The reviewer suggests the following improvements:

1.       The detailed description of the sweet potato cultivar and the information on if it is susceptible or resistant to F. solani infection. This information will be important to interpret the different DEGs data from the previous report on F. oxysporum.

2.       The F. solani strain used for inoculation also needs to be molecularly identified, preferably with its barcoding region sequence deposited with GenBank.

3.       Table 2. T-6h and T24h data had very low mapped percentages to sweet potato genome (10.46-13.46%), dramatically different from the CK, T3d and T-5d data. This is quite unexpected. The authors suggest that “so the pathogen is very active in infecting sweetpotato within 24 hr”. If the authors suggest more transcript contribution by F. solani, then these data should mostly be mapped to F. solani genome. The authors need to verify this point.

4.       There is the need to discuss if the previously identified resistance genes (those described in the introduction part, also those identified by the previous paper (Lin et al. 2017) to Fusarium infection) were among the DEGs. If not, there should be some discussions.

5.       The English language quality needs to be improved. There are grammar errors here and there and many words are confusing, making it difficult for the reader to follow the writer’s intended meaning. I give some examples below, with the problematic wording underlined:

4a) lane50-51: These genes were related to the mitogen-activated protein kinase (MAPK), TFs, glutathione transferases (GSTs) catalyze cascade respons [12-15].

4b) lane 238: “regulated genes in calcium-dependent protein kinase in CK-32 vs. T-6h, CK-32 vs. T-24h”, should this be ‘genes for calcium dependent protein kinase’?

4c) lane 246-247: “In calcium channel, except genes about calcium-dependent protein kinase and respiratory burst oxidase, there were other four types of genes…”. Do you mean “in addition to”?

Author Response

Dear professor,

Thank you very much for your time involved in reviewing the manuscript. We are so grateful for your clear and detailed feedback and hope that the explanation has fully addressed all of your concerns. We discuss each of your comments individually along with our corresponding responses. Please see the attachment.

Best regards,

   Chengling Zhang

Reviewer 2 Report

The MS Transcriptome Characterization and Genes Changes in the Sweetpotato Foots Induced by Fusarium Solani investigate an important topic however it needs a major overhaul before publications.

1.     In title “s”olani should be small.

2.     Abstract should describe the background of the study, why it needed along with recommendations.

3.     Line 34 Fusarium solani full

4.     Line 67-70, please describe clear novelty of this work. Earlier published work on the same topic and how your work in novel than earlier published work.

5.     Line 82 Fusarium full please follow same trend where sentence is stating with F. solan, keep Fusarium full.

6.     How did you confirm that it was Fusarium solani, did you do sequencing? morphological features.

7.     Petri plate should be capital.  

8.     On what basis you determined the disease severity, please mention in the MS.

9.     While inoculation, where did you do this experiment, what was the relative humidity, which variety of sweet potato you used, was it susceptible to Fusarium, these all details are required in materials and methods. Before conducting this experiment, did you do pathogenicity test?

10.  The heat map can be given as supplementary file.

11.  Can you give the image of treated and control sweet potato with symptoms that developed during experiments.

12.  Discussion looks very weak please discuss with more literature.

Author Response

Dear professor,

Thank you very much for your time involved in reviewing the manuscript. We are so grateful for your clear and detailed feedback and hope that the explanation has fully addressed all of your concerns. We discuss each of your comments individually along with our corresponding responses. Please see the attachment.

Best regards

Round 2

Reviewer 1 Report

The many changes according to reviewers' suggestions and comments are noted. As a result, the manuscript has been significantly improved. Ideally, a resistant strain should be analyzed as well, preferably together with a susceptible strain. Such a comparative study will be very helpful to shortlist the true resistant genes. 

For the response to point 2, the authors claim to have barcoded the F. solani strains and deposited them with GenBank. This is no mention of this, nor the inclusion of the accession numbers in the revised manuscript.  

Author Response

Dear professor,

Many thanks again for your quick reply and constructive comments concerning our manuscript entitled “Transcriptome Characterization and Gene Changes in the Sweetpotato Roots Induced by Fusarium solani” for the “genes”.

According to your suggestion, we have carefully revised the MS. Please see the attachment.

Best regards,

  Chengling  Zhang
